# Transient risk of ambient fine particulate matter on hourly cardiovascular events in Tainan City, Taiwan

Pei-Chih Wu[1,2], Tain-Junn Cheng[3]*, Cheng-Pin Kuo[4], Joshua S. Fu[4], Hsin-Chih Lai[1], Tsu-Yun Chiu[5], Li-Wei Lai[5]

**1** Department of Green Energy and Environmental Resources, Chang Jung Christian University, Tainan, Taiwan, **2** Department of Occupational and Safety and Health, Chang Jung Christian University, Tainan, Taiwan, **3** Departments of Neurology and Occupational Medicine, Chi Mei Medical Center, Tainan, Taiwan, **4** Department of Civil and Environmental Engineering, University of Tennessee Knoxville, Knoxville, Tennessee, United States of America, **5** Environmental Research and Information Center, Chang Jung Christian University, Tainan, Taiwan

* tjcheng@mail.chimei.org.tw

**Data Availability Statement:** All relevant data are within the paper and its Supporting Information files.

## Abstract

### Background

The association between daily changes in ambient fine particulate matter ($PM_{2.5}$) and cardiovascular diseases have been well established in mechanistic, epidemiologic and exposure studies. Only a few studies examined the effect of hourly variations in air pollution on triggering cardiovascular events. Whether the current $PM_{2.5}$ standards can protect vulnerable individuals with chronic cardiovascular diseases remain uncertain.

### Methods

we conducted a time-stratified, case-crossover study to assess the associations between hourly changes in $PM_{2.5}$ levels and the vascular disease onset in residents of Tainan City, Taiwan, visiting Emergency Room of Chi Mei Medical Center between January 2006 and December 2016. There were 26,749 cases including 10,310 females (38.5%) and 16,439 males (61.5%) identified. The time of emergency visit was identified as the onset for each case and control cases were selected as the same times on other days, on the same day of the week in the same month and year respectively. Residential address was used to identify the ambient air pollution exposure concentrations from the closest station. Conditional logistic regression with the stepwise selection method was used to estimate adjusted odds ratios (ORs) for the association.

### Results

When we only included cases occurring at $PM_{2.5} > 10$ μg/m$^3$ and $PM_{2.5} > 25$ μg/m$^3$, very significant ORs could be observed for 10 μg/m$^3$ increases in $PM_{2.5}$ at 0 and 1 hour, implying fine particulate exposure could promptly trigger vascular disease events. Moreover, a very clear increase in risk could be observed with cumulative exposure from 0 to 48 hours, especially in those cases where $PM_{2.5} > 25$ μg/m$^3$.

**Funding:** This study was supported by the grants from Chi Mei Medical Center (CMFHR10752, 2018) and Environmental Protection Administration, ROC (Taiwan) (EPA-106-FA18-03-A260, 2017). The funders had no role in study design, data collection and analysis, decision to publish, or preparation of the manuscript.

**Competing interests:** The authors have declared that no competing interests exist.

## Conclusions

Our study demonstrated that transient and low concentrations of ambient $PM_{2.5}$ trigger adult vascular disease events, especially cerebrovascular disease, regardless of age, sex, and exposure timing. Warning and delivery systems should be setup to protect people from these prompt adverse health impacts.

## Introduction

Cardiovascular diseases (CVDs) are leading causes of death in the world in recent decades. Global burden of diseases in 2015 also clearly showed that CVDs remain a major cause of health losses for all regions of the world. Although a dramatic decline in CVD mortality has been seen in regions with very high sociodemographic indexes, but only a gradual decrease or no change has occurred in most regions [1]. In these regions, ambient $PM_{2.5}$ contributed the greatest burdens of diseases, which increased over the past 25 years, due to aging population and increasing air pollution in low-income and middle-income countries [2]. Numerous studies have provided persuasive evidence that short-term exposure to $PM_{2.5}$ over a period of a few hours to weeks can trigger CVD-related mortality and non-fatal events, especially among susceptible individuals. Susceptible individuals at greater risk may include elderly and obese patients with preexisting coronary artery disease, and chronic metabolic diseases. A systemic review of the American Heart Association (AHA) also specifically demonstrated that the $PM_{2.5}$ concentration-cardiovascular risk relationships for both short- and long-term exposure tend to fall below 15 ug/m$^3$ without a recognizable "safe" threshold. Many biological mechanisms have shown PM exposure could exacerbate existing CVDs and trigger acute cardiovascular events (within a few hours) and also addressed the need to investigate ultra-acute peak PM excursions (e.g., 1 to 2 hours), which have been less studied. The time frame of awareness and personal protection in susceptible subjects should be considered in clinical recommendation [3].

Many epidemiologic studies have well addressed the concentration-relationship between short-term exposure to $PM_{2.5}$ and cardiovascular events, including ischemic heart disease, heart failure, cardiac arrhythmia/arrest, and ischemic strokes in daily time scales of air pollution. However, only a small number of epidemiological studies have examined the effects of hourly variations in air pollution on triggering CVD events. Intracerebral hemorrhage mortality has been found to be associated with the hourly concentration of suspended particulate matter (1–2 hour lag), according to the Ministry of Health, Labour, and Welfare of Japan [4]. A study conducted in Okayama, Japan also found that suspended particulate matter exposure of 0 to <6 hours before the case events was significantly associated with risks of onset of cardiovascular and cerebrovascular disease in emergency hospital visits [5]. One study conducted in the Boston area further found that the increase in risk of ischemic stroke onset was greatest within 12 to 14 hours of exposure to ambient fine particulate matter air pollution [6]. However, studies conducted in 8 cities in Ontario [7], Houston [8], and South London [9] do not support the hypothesis that hourly increases in $PM_{2.5}$ levels are associated with CVD events.

Our study used a time-stratified, case-crossover design to assess the associations between hourly changes in $PM_{2.5}$ levels and the risk of cardiovascular disease onset in residents of Tainan City, Taiwan, who had visited emergency rooms in Chi Mei Medical Center between January 2006 and December 2016.

## Materials and methods

### Study design and subjects

A time-stratified, case-crossover design was used in this study [10]. A case-crossover study can be thought of as a type of self-matched, case control study. For each individual case, exposure before the event (case period) is compared to exposure at other control periods. Hourly hospital admission data (2006–2016) were collected from Chi Mei Medical Center in Tainan City. Patients with cardiovascular diseases (N = 26749) aged $\geq$20 years old were selected according to the International Classification of Diseases 9th edition, Clinical Modification (ICD-9 CM) codes (428 Heart failure, 426–427 Arrhythmia, 430–438 Cerebrovascular disease, 410–414 and 429 Ischemic heart disease, 440–449 Peripheral vascular disease). We used the time of emergency visit as the onset for each case. Control periods for each case were selected as the same times on other days, on the same days of the week in the same months and years [11]. Residential addresses were used to identify the ambient air pollution exposure concentrations from the closest station.

### Air pollution data and meteorological data

We obtained hourly pollutant concentrations of $PM_{10}$, $PM_{2.5}$, $SO_2$, $NO_2$, NO, $O_3$, CO, total hydrocarbons (THC), and non-methane hydrocarbons (NMHC), and meteorological indices of temperature, relative humidity, and wind speed at 4 air quality monitoring stations (AQMS) of the Taiwan EPA in Tainan City during the study period. Hourly data for each case and control period were collected from the closest station.

### Statistical analyses

Conditional logistic regression with the stepwise selection method was used to estimate adjusted odds ratios (ORs) and 95% confidence intervals (CIs) for the association between air pollutant exposure and emergency visits due to cardiovascular diseases. To fully adjust for the potential time-variant confounders, we used natural cubic splines with 3 degrees of freedom (df) for the ambient temperature, relative humidity and wind speed in all models. All case event periods and cases event periods in $PM_{2.5}>10$ μg/m$^3$, $PM_{2.5}>25$ μg/m$^3$, and $PM_{2.5}>54$ μg/m$^3$ were also identified to assess the effects per 10 μg/m$^3$ increases in $PM_{2.5}$. In all analyses, we tested the linearity assumption between air pollution exposure and health outcomes by replacing the continuous exposure variables by a natural spline with 3 degrees of freedom and compared the model fit by the likelihood ratio test. Goodness of fit among different models was assessed by AIC. Collinearity among variables was assessed by VIF. We conducted analyses using the SAS statistical package. This study is approved by the Institutional Review Board of Chi Mei Medical Center on 14th Dec. 2018 (No.10612-012) and is exempt from obtaining inform consent.

## Results

All the patients visiting Chi Mei Medical Center with defined vascular disease events were enrolled from Jan. 2006 to Dec. 2016. There were 26,749 cases including 10,310 females (38.5%) and 16,439 males (61.5%) identified during the study period. To explore the threshold effects of different fine particulate matter concentrations, we used 24 hours average $PM_{2.5}$ levels prior to events to classify cases in different exposure backgrounds, 10 μg/m$^3$ (WHO annual mean guideline value) and 25 μg/m$^3$ (WHO 24-hour mean guideline value). We identified 21,756 cardiovascular cases with $PM_{2.5}>10$ μg/m$^3$ including 8409 females (38.7%) and 13,347 males (61.3%), and 12,524 cases were identified with $PM_{2.5}>25$ μg/m$^3$ including 4,894 females

**Table 1. Characteristics of hospital emergency visits for patients with cardiovascular disease in a medical center in Tainan City, 2006 to 2016 in different PM$_{2.5}$ ambient backgrounds (24 hours average before emergency visits).**

| | All cases | | PM$_{2.5}$>10 µg/m$^3$ | | PM$_{2.5}$>25 µg/m$^3$ | |
|---|---|---|---|---|---|---|
| | N | (%) | N | (%) | N | (%) |
| **Total** | 26749 | (100.0%) | 21756 | (100.0%) | 12524 | (100.0%) |
| **Gender** | | | | | | |
| Female | 10310 | (38.5%) | 8409 | (38.7%) | 4894 | (39.1%) |
| Male | 16439 | (61.5%) | 13347 | (61.3%) | 7630 | (60.9%) |
| **Age** | | | | | | |
| < 65 | 12157 | (45.4%) | 9795 | (45.0%) | 5633 | (45.0%) |
| ≥ 65 | 14592 | (54.6%) | 11961 | (55.0%) | 6891 | (55.0%) |
| **Onset time** | | | | | | |
| Daytime (8 am- 7 pm) | 9155 | (34.2%) | 7433 | (34.2%) | 4255 | (34.0%) |
| Nighttime (8 pm- 7 am) | 17594 | (65.8%) | 14323 | (65.8%) | 8269 | (66.0%) |
| **Cardiovascular disease** | | | | | | |
| Heart failure | 2127 | (8.0%) | 1760 | (8.1%) | 1055 | (8.4%) |
| Arrhythmia | 2280 | (8.5%) | 1871 | (8.6%) | 1097 | (8.8%) |
| Cerebrovascular disease | 15322 | (57.3%) | 12345 | (56.7%) | 7093 | (56.6%) |
| Ischemic heart disease | 5918 | (22.1%) | 4881 | (22.4%) | 2740 | (21.9%) |
| Peripheral vascular disease | 1102 | (4.1%) | 899 | (4.1%) | 539 | (4.3%) |

(39.1%) and 7,630 males (60.9%) (Table 1). The demographic characteristics of cases among different background levels were almost constant. Briefly, the patients are predominantly male (60.9–61.5%) and elders (54.6–55.0%), with higher rates of nighttime attacks (65.8–66.0%). Among the cardiovascular diseases, cerebrovascular disease (56.6–57.3%) is the major disease attack, followed by ischemic heart disease (21.9–22.4%), arrhythmia (8.5–8.8%), heart failure (8.0–8.4%) and peripheral vascular disease (4.1–4.3%).

The average air pollutants and meteorological data in both control periods and case periods are presented in Table 2. The average temperature degree (24.71 V.S. 21.14) and relative humidity percentage (75.86% V.S. 68.57%) are higher in control periods. The average wind speed velocity (2.49 m/s V.S. 2.36 m/s) is faster in case periods. The average concentration of pollutants of PM$_{10}$ (126.98 µg/m$^3$ V.S. 67.37 µg/m$^3$), PM$_{2.5}$ (48.6 µg/m$^3$ V.S. 25.8 µg/m$^3$), NO$_2$

**Table 2. Twenty-four hours average air pollutants and meteorological data in case periods and control periods.**

| Species | Unit | Control periods | | Case periods | |
|---|---|---|---|---|---|
| | | Mean | SD | Mean | SD |
| **Temperature** | degree | 24.71 | 5.22 | 21.14 | 5.01 |
| **Relative Humidity** | % | 75.86 | 11.25 | 68.57 | 13.25 |
| **Wind Speed** | m/s | 2.36 | 1.32 | 2.49 | 1.39 |
| **PM$_{10}$** | µg/m$^3$ | 67.37 | 39.01 | 126.98 | 97.66 |
| **PM$_{2.5}$** | µg/m$^3$ | 25.8 | 18.61 | 48.6 | 39.6 |
| **NO$_2$** | ppb | 14.49 | 8.34 | 19.69 | 8.92 |
| **NO** | ppb | 3.37 | 4.58 | 3.64 | 5.01 |
| **SO$_2$** | ppb | 3.77 | 2.23 | 5.4 | 2.7 |
| **CO** | ppm | 0.4 | 0.2 | 0.54 | 0.2 |
| **O$_3$** | ppb | 30.15 | 22.37 | 36.39 | 25.05 |

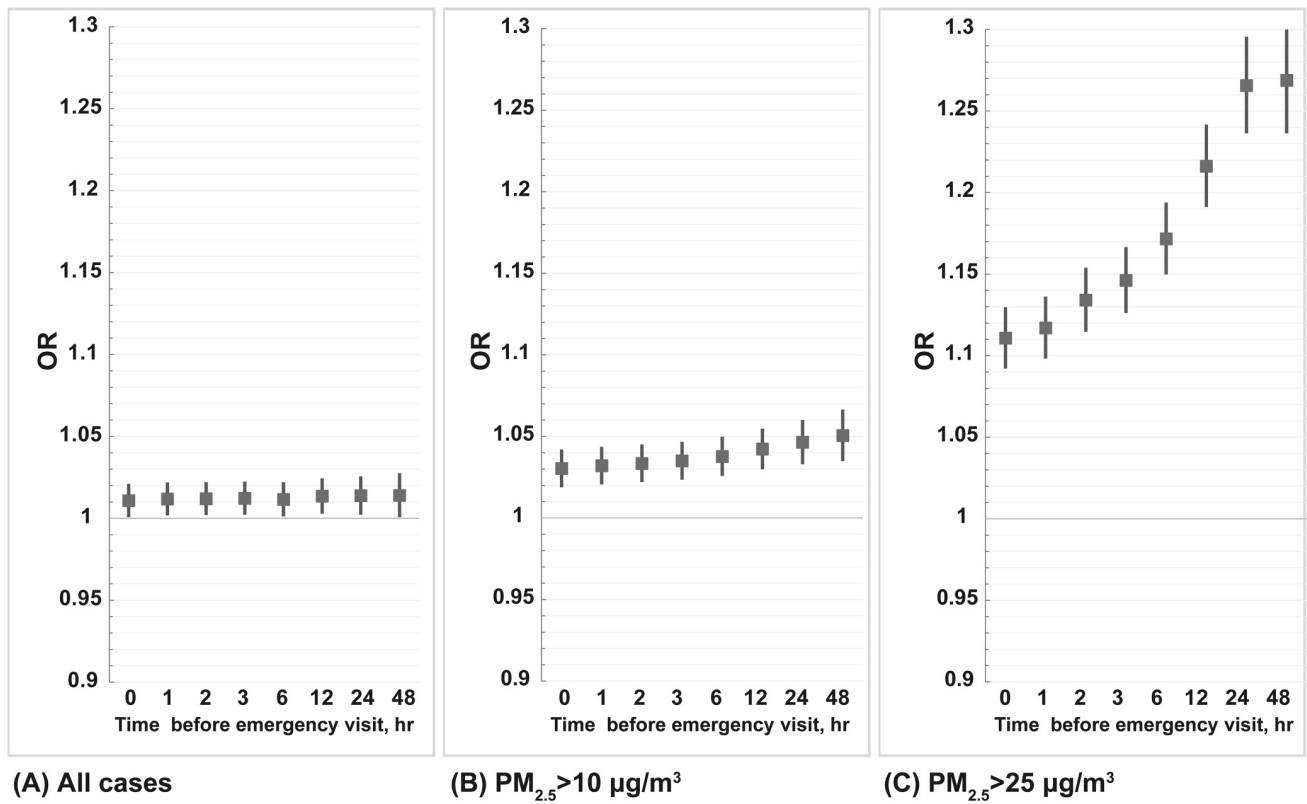

**Fig 1. Odds ratio of cardiovascular emergency visits for 10 µg/m³ increase in PM2.5 based on the time weighted average before disease onset by using single-pollutant logistic regression in different PM2.5 case periods.**

(19.69 ppb V.S. 14.49 ppb), NO (3.64 ppb V.S. 3.37 ppb), SO₂ (5.4 ppb V.S. 30.15 ppb), CO (0.54 ppb V.S. 0.4 ppb), and O₃ (36.39 ppb V.S. 14.49 ppb) are all higher in case periods.

The odds ratio (OR) of vascular disease events and PM2.5 was first assessed by a single-pollutant model (Fig 1). Odds ratios of cardiovascular emergency visits with 10 µg/m³ increases in PM2.5 at 0 to 48 hours before disease onset were all statistically significant when including all cases in the model.

The lag–response relationship increased slightly with cumulative exposure before the case event from 0 to 48 hours in all cases. When we only include cases occurring with PM2.5>10 µg/m³ and PM2.5>25 µg/m³, very significant ORs could be observed for 10 µg/m³ increases in PM2.5 at 0 and 1 hour, implying fine particulate exposure could promptly trigger vascular disease events. Moreover, a very clear increased risk level could be observed with cumulative exposure from 0 to 48 hours, especially in those cases identified when PM2.5>25 µg/m³.

The linear model and non-linear model were assessed under the linearity assumption and goodness of fit between air pollution exposure and health outcomes. In comparison with the non-linear model, the linear model presented a better goodness of fit and collinearity in the relationship between OR and 1-hour PM2.5 concentration before emergency visits due to vascular events (Fig 2). Considering applicability for air quality management, the linear model for cardiovascular disease is used to build a multi-pollutant model. Multivariate logistic regression model was used to examine the short-term effects (1 hour exposure before emergency visit) of PM2.5 and emergency visits in cases occurring with PM2.5>10 µg/m³ and PM2.5>25 µg/m³. Adjusted OR per 10 µg/m³ increase in PM2.5 at 1 hour before the emergency visit was higher

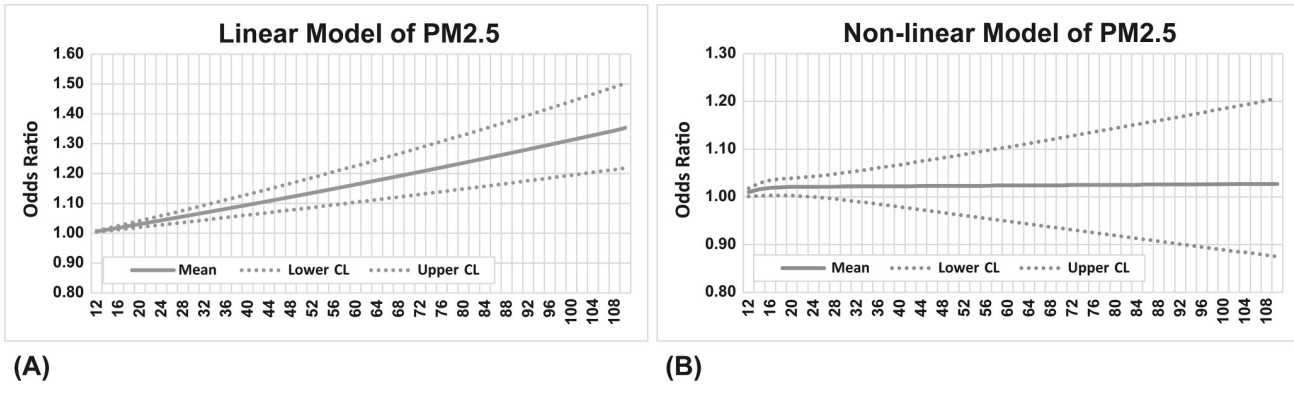

**Fig 2. Relationship between odds ratio and 1-hr PM$_{2.5}$ concentrations before emergency visits.** (A) Linear model, (B) Non-linear model.

in PM$_{2.5}$>25 μg/m$^3$ (OR = 1.06, 95% C.I. 1.03–1.09) compared to those cases occurring at PM$_{2.5}$>10 μg/m$^3$ (OR = 1.03, 95% C.I. 1.02–1.04). Stratifying onset of those cases at daytime and nighttime by using multivariate models showed higher ORs per 10 μg/m$^3$ increase in PM$_{2.5}$ in cases onset at nighttime both could be found when PM$_{2.5}$>10 μg/m$^3$ and PM$_{2.5}$>25 μg/m$^3$. The ORs of vascular disease statistically increased in both age groups (age≧65 and age<65), for males, and for daytime and nighttime admission when PM$_{2.5}$>10 μg/m$^3$ (Fig 3A). Although the ORs increased when PM$_{2.5}$>25 μg/m$^3$ after adjustment, the risk only significantly increased in age≧65 y/o, males, females, and for night time admissions (Fig 3B). The adjusted ORs per 10 μg/m$^3$ increase in PM$_{2.5}$ of different vascular diseases were also examined. After adjustment of multiple pollutants and meteorological indices, risk of cerebrovascular disease onset increased constantly and significantly for both PM$_{2.5}$>10 μg/m$^3$ and PM$_{2.5}$>25 μg/m$^3$ (Fig 4).

## Discussion

Our study results show that the OR of vascular event onset was elevated immediately with current hour PM$_{2.5}$ levels, implying fine particulate exposure could promptly trigger vascular disease event. We also observed that elevated risk persisted from 0 to 48 hours in both PM$_{2.5}$ thresholds of the WHO long-term guideline value 10 μg/m$^3$ and the WHO 24-hour mean guideline value of 25 μg/m$^3$. The relationship between risk of onset of vascular event and short-term PM$_{2.5}$ levels tend to reveal a linear concentration response function. Compared to other type of vascular diseases, cerebrovascular disease was more strongly associated with PM$_{2.5}$ levels in Tainan City.

The transient response of elevated risk of fine particles (PM$_{2.5}$) to vascular disease events in our study was consistent with previous studies applying hourly data analysis. Our study not only shows the transient risk of PM$_{2.5}$ in triggering vascular disease events, but also indicates continuously elevated risk for cumulative exposure of 48 hours. A study conducted in the Boston area showed that hourly PM$_{2.5}$ was associated with transient risk of acute myocardial infarction (AMI) onset (2 hours before onset). An elevated risk of MI was also found in 24-hour average PM$_{2.5}$ concentrations [12]. Another study also conducted in the Boston area found that OR of acute cerebrovascular diseases and ischemic strokes elevated immediately, and peaked in association with mean PM$_{2.5}$ levels 12 to 14 hours earlier and decreased thereafter [6]. A study conducted in southern Germany using traffic exposure as an indicator of fine particle and other traffic-related pollution also found the highest risk at onset of AMI within

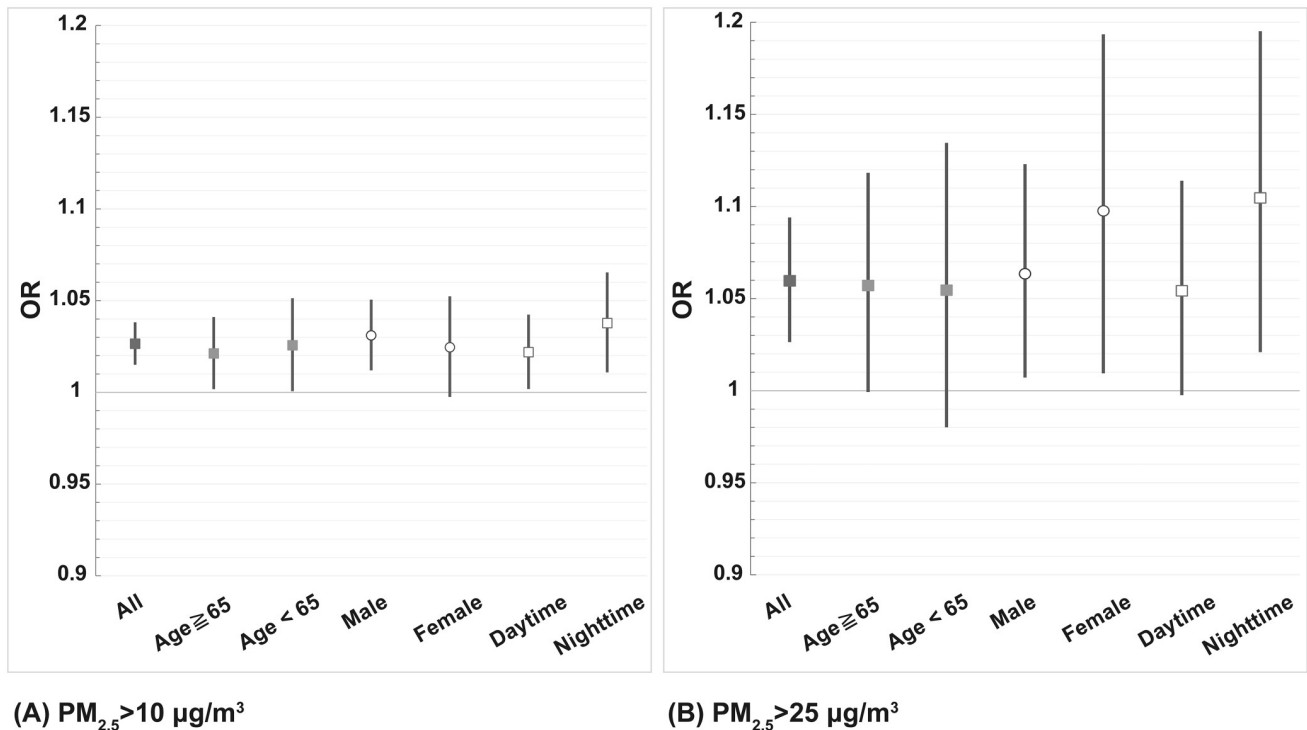

**(A) PM₂.₅>10 μg/m³**

**(B) PM₂.₅>25 μg/m³**

**Fig 3. Adjusted ORs per 10 μg/m³ increase in PM₂.₅ at 1 hour before emergency visits for different ages, genders and periods of emergency visits.** (A) PM₂.₅>10 μg/m³ with model adjusted for ambient temperature, humidity, wind speed, and ozone; (B) PM₂.₅>25 μg/m³ with model adjusted for ambient temperature, relative humidity, wind speed, PM₁₀, CO, and ozone.

one hour after traffic exposure [13]. A further study also demonstrated that elevated risk persisted for up to 6 hours. The data suggested that transient exposure to traffic regardless of the means of transportation may increase the risk of AMI transiently. One study analyzing data of acute coronary syndrome patients treated at the University of Rochester Medical Center (URMC) Cardiac Catheterization Laboratory, with ambient air pollutant concentrations measured in Rochester, New York, by case-crossover methods, also found significant increases in the risk of acute coronary syndromes associated with each 7.1 μg/m³ increase of PM₂.₅ concentration in the 1 hour prior to acute coronary syndrome attack [14]. These increase in transient risks immediately after fine particle exposure are supported by many human exposure studies and these findings demonstrate that inhalation of ambient PM₂.₅ causes rapid changes in nocturnal heart rate variability (25 minutes exposure) and acute conduit artery vasoconstriction (2 hours inhalation) [15, 16]. Experimental studies testing diesel exhaust inhalation have also demonstrated acute vascular dysfunction, ischemia, and thrombotic dysfunction within a few hours of exposure [17–20]. Our findings confirm that many potential biological mechanisms of PM₂.₅ exposure could exacerbate existing CVDs and trigger acute vascular events. Higher ORs could be observed in those cases with event onset at night in our study. This further supports the view that a few hours of cumulative exposure during daytime (with higher traffic and industrial activities) increases the risk of triggering vascular events night.

Lisabeth et al. studied the data from the population-based Brain Attack Surveillance in Corpus Christi (BASIC) Project and found a borderline significant association between one-day PM₂.₅ and O₃ exposure and ischemic stroke/TIA risk even in a community with relatively low pollutant levels with median PM₂.₅ of 7.0 μg/m³ [21]. O'Donnell et al. conducted a study

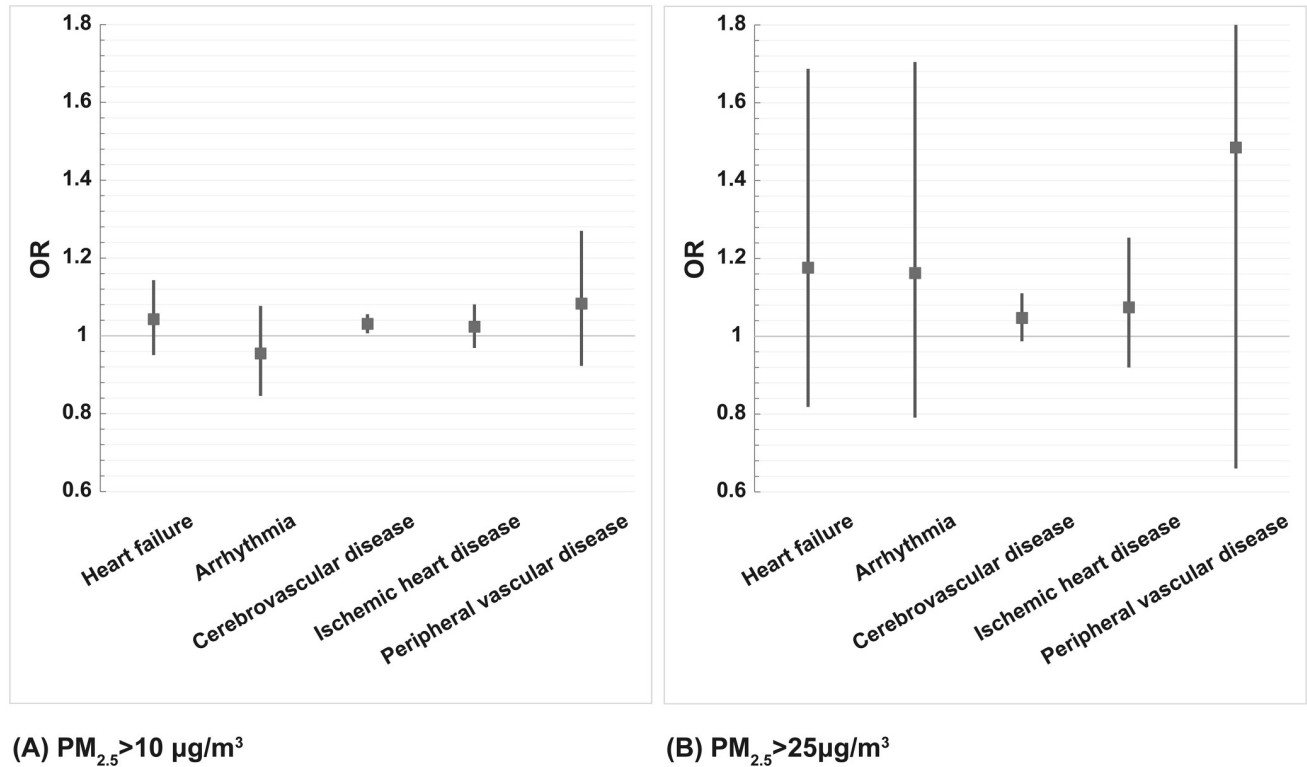

**(A) PM$_{2.5}$>10 µg/m$^3$**

**(B) PM$_{2.5}$>25µg/m$^3$**

**Fig 4. Adjusted ORs per 10 µg/m$^3$ increase in PM$_{2.5}$ at 1 hour before emergency visits in different types of cardiovascular diseases.** (A) PM$_{2.5}$>10 µg/m$^3$ with model adjusted for ambient temperature, humidity, wind speed, and ozone, (B) PM$_{2.5}$>25 µg/m$^3$ with model adjusted for ambient temperature, relative humidity, wind speed, PM$_{10}$, CO, and ozone.

Ontario, Canada, but did not find the association between short-term increase in PM$_{2.5}$ levels and ischemic stroke risk overall by using the average of the values available from all stations by hour in studied areas. However, they found the association between PM$_{2.5}$ and ischemic stroke risk due to large artery atherosclerosis and small vessel disease [7]. Our study demonstrated that cerebrovascular disease accounts for the largest portion and most significant vascular disease event after PM$_{2.5}$ exposure. In comparison with the study mentioned above, our study is more informative because we applied hourly pollutant concentrations and meteorological indices from the station nearest to the event case, which provides more representative exposure data than the average data applied in O'Donnell's study and higher ambient PM$_{2.5}$ exposure concentration of 10 µg/m$^3$ than that in Lisabeth's study.

Based on the findings from previous studies and our study, cerebrovascular disease is the most significant associated vascular disease in short-term and low concentrations of ambient PM$_{2.5}$ exposure. This phenomenon was found especially in large artery atherosclerosis and small vessel disease stroke patients [7]. Besides atherosclerosis, PM$_{2.5}$ probably could also trigger cerebrovascular disease by small vessel occlusion mechanisms. Some plausible mechanisms have been studied and reviewed. O'Neill et al. found that ambient PM$_{2.5}$ exposure is positively associated with vascular cell adhesion molecule 1 (VCAM-1) in type II Diabetes Mellitus patients, suggesting inflammatory response in vascular disease development [22]. Wellenius et al. found that exposure to PM$_{2.5}$ with 3.0 µg/m$^3$ interquartile range increase was associated with higher resting cerebrovascular resistance and lower cerebral blood flow velocity without significant change in resting mean arterial pressure [23]. Furthermore, these changes did not

differ significantly among participants with diabetes mellitus, hypertension, smoking, or by sex or season. These results were compatible with our findings that risk increases even with minimal ambient PM$_{2.5}$ level increases, regardless of sex or age.

There are some vascular effects after short-term ambient air pollution exposure which could be possibly correlated with the cerebrovascular vessels including raising arterial blood pressure, increased vascular resistance, and decreased small-vessel elasticity [5, 7, 21, 24–27]. These findings could be related to attenuation of the endothelium-dependent vascular dilatation response after air pollutant [12, 28] exposure by affecting NO release [29, 30]. Moreover, the resting cerebral blood flow velocity is lower in patients with diabetes mellitus microvascular complications and has been shown to be correlated with systemic arterial stiffness [6, 31]. Lower cerebral blood flow velocity and arterial stiffness could lead to risk of vascular occlusion, which in part explains our findings related to stroke risk increment. However, the exact mechanisms need to be further verified in more detailed clinical study.

The systemic review of epidemiological studies of PM$_{2.5}$ concentration-cardiovascular risk relationships for both short- and long-term exposures points a monotonic increasing function. The elevated risk was found to extend below 15 μg/m$^3$ without a discernable "safe" threshold [3]. Our study found that the association between PM$_{2.5}$ level 1 hour before emergency visit and risk of vascular disease onset showed a monotonic linear function. This finding was also found in the Boston area according to US National Ambient Air Quality Standards, with approximately linear association between PM$_{2.5}$ level and risk of ischemic stroke onset. These results suggest that PM$_{2.5}$ exposure increases the risk of ischemic stroke event sat levels below those currently considered safe under US regulations [6]. The US, Taiwan, and many countries of the world have standard regulations for 24-hour and annual average concentrations and do not address transient elevations (minutes to hours) in fine-particle concentration. Our study further used WHO air quality guideline values, 10 μg/m$^3$ (annual mean) and 25 μg/m$^3$ (24-hour mean), to select our cases for different background levels to estimate the time course of the association between PM$_{2.5}$ levels and vascular disease event onset. Immediate increases in OR for vascular events and elevation of OR following cumulative exposure could even be observed in those cases using the 10 μg/m$^3$ threshold, which implies the safe threshold of transient exposure for vascular disease events could fall below 15 μg/m$^3$. This finding points to the importance of implementing various strategies to reducing daily individual exposures, especially for those populations at high risk (e.g., the elderly, and individuals with preexisting coronary artery diseases, metabolic syndrome or multiple risk factors, or diabetes). Moreover, public health benefits could be gained from lowering PM$_{2.5}$ concentrations even below the present common standards.

Ambient levels of fine particulate vary seasonally, day to day, at different times of day, and in different microenvironments. Studies conducted in urban building environment also demonstrate vertical and horizontal differences [32, 33]. Individuals have to be able to anticipate when and where pollutant levels are likely to vary in order to mitigate personal health risks. An early warning system or real-time alerts of transient elevations in fine particle levels for protecting the most susceptible population could be considered as national or regional adaptations for public health. Multi-media alerts could also facilitate communication by using mobile phones and personal wearable technology in current time. A study conducted in Canada evaluating the impacts of a phone warning and advising system for individuals vulnerable to smog showed that they are more likely to adopt the recommended behaviors than are non-vulnerable individuals [34]. An online survey investigated the frequency at which American health professionals talk about limiting air pollution exposure for patients with respiratory or cardiovascular diseases. Only 34% of respondents, corresponding to 16% of overall health providers, reported talking with their patients with respiratory or cardiovascular disease diagnoses. This

suggests a high degree of opportunities for improving awareness about strategies to limit air pollution exposure among vulnerable individuals and their healthcare providers [35]. Media alerts of the air quality index have long been recognized as an effective strategy to inform people with asthma and the general public to reduce or change their outdoor activities and thus avoid unhealthy or harmful effects [36, 37]. For better understanding awareness of air quality alerts and specific actions taken, especially by at-risk individuals, a Consumer Styles survey was conducted to analyze 12,599 US adults with self-reported asthma, emphysema/chronic obstructive pulmonary disease, and heart disease, for each aspect of air quality awareness. Respondents with respiratory diseases showed higher levels of air quality awareness and behavior to reduce air pollution exposure. However, patients with heart diseases were not associated with higher levels of air quality awareness. These findings suggest important opportunities to raise the awareness of air quality alerts and promote behavior changes to reduce air pollution exposure, especially for those patients with CVD and also patients without CVD who are at high risk [35]. After reviewing 21 studies, D'Antoni found that actual adherence to health advice accompanying air quality warning systems was associated with several psychosocial factors such as knowledge on where to check air quality indices, beliefs that one's symptoms were due to air pollution, perceived severity of air pollution, and advice received from health care professionals [38]. Prevision and warning information delivery to make people aware and positive responses to potential health events should be emphasized along with technical and complex information collection methods [39]. Most of the ambient pollutant warning systems offering health advice according to epidemiological evidence are categorized by levels, indexes, or bands, which vary from country or country to offer. From the experiences of the Air Pollution Warning System in Quebec, the thresholds of warning systems vary in different areas and seasons according to the mortality data and need to be verified [40]. Recently, a newly recommended AQI was lunched in the United Kingdom [39]. According this AQI, a 'Low' band suggests that 24 hour PM$_{2.5}$ levels of 1–3 (0–11 μg/m$^3$, 12–23 μg/m$^3$ and 24–35 μg/m$^3$) are unlikely to lead to adverse effects of short-term exposure. Both susceptible and general people can enjoy their activities as usual. The susceptible adults and children with lung and cardiac problems with symptoms should reduce their activities, especially outdoors, when the 24 hour PM$_{2.5}$ level $> = 36$ μg/m$^3$ reaches the moderate band/index 4 or above. The ambient air quality monitoring systems are well set up in Taiwan. The monitoring system presents real-time AQI from local AQIM stations and forecasts daily AQI to the public for reference. Policies are in place to monitor programs and continuously revise guidance for local and national regulatory policies by regularly monitoring health events, providing warnings and setting up delivery systems to instruct vulnerable people to avoid high concentrations of ambient PM$_{2.5}$ exposure so as to prevent vascular disease events. According to our study results, the risk of vascular disease events increases at levels as low as PM$_{2.5}$ $>10$ μg/m$^3$ with short time lag of just one hour after ambient PM$_{2.5}$ exposure. We strongly recommend that susceptible people be made aware of the potential risk of acute vascular events and take protection action when the ambient PM$_{2.5}$ $>10$ μg/m$^3$, such as tight blood pressure control, and good medication compliance.

In summary, our study demonstrated that transient and low concentrations of ambient PM$_{2.5}$ trigger adult vascular disease events, especially cerebrovascular disease, regardless of age, sex, and exposure timing. Warning and delivery systems should be set up to protect people from these prompt adverse health impacts.

## Supporting information

**S1 Data.**
(XLSX)

## Author Contributions

**Conceptualization:** Pei-Chih Wu, Tain-Junn Cheng.

**Data curation:** Tsu-Yun Chiu.

**Formal analysis:** Cheng-Pin Kuo.

**Funding acquisition:** Pei-Chih Wu, Tain-Junn Cheng.

**Investigation:** Tain-Junn Cheng.

**Methodology:** Pei-Chih Wu.

**Resources:** Hsin-Chih Lai.

**Software:** Cheng-Pin Kuo, Tsu-Yun Chiu.

**Supervision:** Joshua S. Fu.

**Validation:** Hsin-Chih Lai.

**Visualization:** Li-Wei Lai.

**Writing – original draft:** Pei-Chih Wu, Tain-Junn Cheng.

**Writing – review & editing:** Tain-Junn Cheng.

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
