## [Decision Letter · Decision Letter 0]

28 Jul 2020

PONE-D-20-18170

Transient risk of ambient fine particulate matter on hourly cardiovascular events in Tainan City, Taiwan

PLOS ONE

Dear Dr. Cheng,

Thank you for submitting your manuscript to PLOS ONE. After careful consideration, we feel that it has merit but does not fully meet PLOS ONE’s publication criteria as it currently stands. Therefore, we invite you to submit a revised version of the manuscript that addresses the points raised during the review process.

We look forward to receiving your revised manuscript.

Kind regards,

Maria Alessandra Ragusa, PhD Professor

Academic Editor

PLOS ONE

Additional Editor Comments:

The paper, after minor revision according to the attached report, could be accepted.

Journal Requirements:

Reviewers' comments:

Reviewer's Responses to Questions

**Comments to the Author**

1. Is the manuscript technically sound, and do the data support the conclusions?

Reviewer #1: Yes

2. Has the statistical analysis been performed appropriately and rigorously? 

Reviewer #1: Yes

3. Have the authors made all data underlying the findings in their manuscript fully available?

Reviewer #1: Yes

4. Is the manuscript presented in an intelligible fashion and written in standard English?

Reviewer #1: Yes

5. Review Comments to the Author

Reviewer #1: I found some merits in this methodology, the results obtained are interesting and deserve to be disclosed, as they could provide useful information to policymakers. The topic exposed is in compliance with the arguments required in this journal and the format is respected too. The statistical analysis performed is good and I find the application in this field original. However, I suggest improving the environmental aspect of air quality in big cities adds some specific works to make this work more complete. I suggest you some papers about it:

-Row 302 add: “Analysis of vertical profile of particulates dispersion in function of the aerodynamic diameter at a

congested road in Catania”

-Row 318 add: “Evaluation of the air pollution in a Mediterranean region by the air quality index”

-Row 326 add: “Air quality data for Catania: analysis and investigation case study 2012-2013”

If you improve the weakness of this theme, this manuscript will deserve to be published in this journal.

6. PLOS authors have the option to publish the peer review history of their article (what does this mean?). If published, this will include your full peer review and any attached files.

Reviewer #1: No

---

## [Author Response · Author response to Decision Letter 0]

7 Aug 2020

<<Response to Journal Requirements>>

1. Please provide additional details regarding participant consent. In the ethics statement in the Methods and online submission information, please ensure that you have specified (1) whether consent was informed and (2) what type you obtained (for instance, written or verbal, and if verbal, how it was documented and witnessed). If your study included minors, state whether you obtained consent from parents or guardians. If the need for consent was waived by the ethics committee, please include this information.

Response: We added the statement about the waiving inform consent approval information “This study is approved by the Institutional Review Board of Chi Mei Medical Center on 14th Dec. 2018 (No.10612-012) and is exempt from obtaining inform consent.” in line 122-4.

<<Response to Reviewer #1>>

1. Reviewer #1: I found some merits in this methodology, the results obtained are interesting and deserve to be disclosed, as they could provide useful information to policymakers. The topic exposed is in compliance with the arguments required in this journal and the format is respected too. The statistical analysis performed is good and I find the application in this field original. However, I suggest improving the environmental aspect of air quality in big cities adds some specific works to make this work more complete. I suggest you some papers about it:

-Row 302 add: “Analysis of vertical profile of particulates dispersion in function of the aerodynamic diameter at acongested road in Catania”

-Row 318 add: “Evaluation of the air pollution in a Mediterranean region by the air quality index”

-Row 326 add: “Air quality data for Catania: analysis and investigation case study 2012-2013”

Response: Thanks for reviewer’s comments and suggestions. We add a sentence “Studies conducted in urban building environment also demonstrate vertical and horizontal differences[32, 33].” at row 303-304 and cite more articles [36, 37] at row 320 to enrich our manuscript.

---

## [Editor Report · Decision Letter 1]

10 Aug 2020

Transient risk of ambient fine particulate matter on hourly cardiovascular events in Tainan City, Taiwan

PONE-D-20-18170R1

Dear Dr. Cheng,

We’re pleased to inform you that your manuscript has been judged scientifically suitable for publication and will be formally accepted for publication once it meets all outstanding technical requirements.

Kind regards,

Maria Alessandra Ragusa, PhD Professor

Academic Editor

PLOS ONE

Additional Editor Comments (optional):

The authors followed all the suggestions and now the paper is ready for publication.
---

## [Editor Report · Acceptance letter]

12 Aug 2020

PONE-D-20-18170R1 

Transient risk of ambient fine particulate matter on hourly cardiovascular events in Tainan City, Taiwan 

Dear Dr. Cheng:

I'm pleased to inform you that your manuscript has been deemed suitable for publication in PLOS ONE. Congratulations! Your manuscript is now with our production department. 

Kind regards, 

on behalf of

Dr. Maria Alessandra Ragusa 

Academic Editor

PLOS ONE